# Mode Optimization of Microelectromechanical-System Traveling-Wave Ultrasonic Motor Based on Kirigami

**DOI:** 10.3390/mi16020239

**Published:** 2025-02-19

**Authors:** Rong Li, Longqi Ran, Cong Wang, Jiangbo He, Wu Zhou

**Affiliations:** 1School of Mechanical Engineering, Xihua University, Chengdu 610039, China; lirong66@stu.xhu.edu.cn (R.L.); 212023085500003@stu.xhu.edu.cn (C.W.); 2School of Mechanical and Electrical Engineering, University of Electronic Science and Technology of China, Chengdu 611731, China; slash_long7@163.com (L.R.); zhouwu916@uestc.edu.cn (W.Z.)

**Keywords:** ultrasonic motor, MEMS, kirigami, optimization

## Abstract

High-quality traveling waves in stators are critical for traveling-wave ultrasonic motors (TUSMs) to achieve good stability and efficiency. However, the modal splitting and shape distortion that is induced by the anisotropic elasticity induce severe traveling wave distortion. In this study, mode optimization based on kirigami is proposed to suppress modal splitting and shape distortion. Initially, the kirigami pattern on the inner boundary of the stator was built by linear interpolation. Subsequently, the optimization model for the orthogonal modes with even and odd nodal diameters was established. An extended Nelder–Mead Simplex Algorithm with the advantages of derivative-free and bound constraints was employed to search the solution. After optimization, the mode shape of the orthogonal modes with odd nodal diameters was much closer to the sine-style. For instance, the distortion of the B13 mode was significantly reduced to 0.003. Meanwhile, the intrinsic frequency matching was still retained after the optimization. In contrast, the optimization suppressed both the frequency splitting and shape distortion of the orthogonal modes, with even nodal diameters. For instance, the frequency splitting relating to the B14 mode was significantly reduced from 380 Hz to 1 Hz, and the shape distortion was as low as 0.004.

## 1. Introduction

Ultrasonic motors have the advantages of a fast response, small size, and negligible electromagnetic interference. They have been widely applied in robotics [1], precision positioning systems [2], medical systems [3], and aerospace vehicles [4].

In recent years, more and more studies have focused on traveling-wave ultrasonic motors (TWUMs), adopting wafer-scale microelectromechanical system (MEMS) fabrication techniques. A MEMS TWUM reduces the size and manufacturing cost, improves manufacturing precision, and enhances the capabilities of system-level integration. Rudy et al. [5,6] proposed a MEMS TWUM with a diameter of 3 mm, wherein the annular stator was mainly made of single-crystal silicon (SCS). Zhou et al. [7] improved the stator quality factor by positioning the supporting beams at the modal shape’s saddle points and introducing a buffering annular frame. Qin et al. [8] proposed an innovative rotor preloading mechanism using magnetic attraction and precise preloading control for the MEMS TWUM. Ran et al. [9] established a three-dimensional contact model to study the actuation characteristics of the MEMS TWUM. Zhao et al. [10] proposed a MEMS TWUM with a thick PZT film, fabricated by electrohydrodynamic jet printing to improve its performance. Yang et al. [11] proposed an embedded capacitive sensor in the MEMS TWUM to detect the rotation angle. Hareesh et al. [12] proposed a MEMS TWUM fabricated from a bulk PZT to solve the problem of residual stress.

It is crucial to excite a high-quality traveling wave on the stator, because a distorted traveling wave degrades the stability and efficiency of a TWUM significantly [13]. However, it is often difficult to avoid modal splitting or shape distortion in practice due to a variety of defect factors. In related research fields, many scholars have conducted in-depth discussions on modal splitting and related issues from different points of view. He et al. [14] proposed that modal splitting can be avoided by adopting orthogonal modes with odd nodal diameters. But the shape distortion still exists. Kumar et al. [15] cleverly used variational methods to derive analytical expressions specifically for predicting the mode splitting caused by teeth. Zhang et al. [16] clearly showed that even if mode splitting still exists, a pure traveling wave can be generated by using pre-selected angular positions. Liu et al. [17] carefully derived the analytical model of modal splitting of annular stators based on the Hamiltonian principle and obtained the elimination rules for modal splitting. S Wang et al. [18] focused on the study of the number of connection characteristics and the modal characteristics of the wave number of the traveling-wave ultrasonic motor with an example cylindrical stator and successfully obtained an effective solution to suppress and eliminate the modal coupling with these two numbers by using the perturbation method. D Zhang et al. [19] used a multi-scale method to obtain the wave distortion data generated by the contraction of PZT and averaged the rotor speed when the torque changed. Rudy et al. [5] studied the mode splitting caused by a tether in the MEMS TWUM thoroughly and proposed an effective strategy of using a flexible tether to suppress the mode splitting. In addition, Zhang et al. [20] also studied the mode splitting of submicron plates, caused by the asymmetry caused by exposure to the surrounding environment.

This work proposes mode optimization based on kirigami to suppress modal splitting and shape distortion simultaneously. In the current research, kirigami, which is based on the cutting of sheet materials, has been successfully employed to modulate the material elasticity and thermal transport characteristics of materials such as graphene and black phosphorus [21,22]. It improves materials by ensuring super elasticity with a small energy loss [23]; therefore, the kirigami pattern was used for the stator and optimized to minimize the objective function referring to the modal splitting and shape distortion.

## 2. Design Principle

### 2.1. Principle of Kirigami

Kirigami is a process technology that changes the shape and structure of a material [24]. The processing of sheet materials by folding or cutting, when stretched or compressed in motion, allows the product to present 2D or 3D patterns and structures, which can increase the elastic limit of the product [23], as shown in Figure 1. Therefore, it is necessary to ensure the continuity and integrity of the product, and the strength and toughness of the material need to be taken into account.

As a process technology for cutting thin-sheet materials, kirigami can create a variety of complex geometric shapes for products to meet diverse design needs [21]. Its production process is usually implemented on a lighter mass, such as a wafer-level structure. Thus, it can be considered for the design of the stator and rotor of ultrasonic motors. In this process, the properties of the material in terms of stiffness and strength will change [25], so the characteristic frequency of the structure can be changed, and this feature can be considered a direction to improve the traveling wave mode. At the same time, kirigami can also reduce the energy loss of some materials when used and improve the elasticity of materials, which is reflected in the research by K. Chen et al. [23]. Compared to other complex manufacturing processes, kirigami is often simpler and can reduce manufacturing costs. Furthermore, it can also be designed to reduce the loss of raw materials.

### 2.2. Kirigami Pattern on Inner Boundary of Stator

This work implements optimization based on kirigami to suppress mode imperfections, including shape distortion and modal splitting.

In this study, an annular stator with a clamped outer boundary was selected to study the traveling wave distortion, as shown in Figure 2a. This type of annular stator has advantages such as the location of electrical contacts and a hollow center for optical purposes compared to the clamped central zone. Additionally, this type of annular stator can withstand much higher preloading than flexible tethers on the outer edge [26].

The kirigami is made on the inner boundary, because the outer boundary is fixed to support the whole stator, as shown in Figure 2a. The kirigami or cutting increases the hole, making the inner boundary no longer a circle. The kirigami pattern creates that boundary after the cutting is symmetric in relation to axes in the 0°, 45°, 90°, and 135° orientations. This kirigami pattern has two crucial advantages. The first one is that the stiffness and mass still have an angle of 90°, and consequently, the intrinsic matching for the orthogonal modes with odd nodal diameters can be retained. The second one is that only the boundary from 0° to 45° needs to be described.

The linear interpolation describes the boundary in this work, as shown in Figure 2b. If the number of interpolation points from 0° to 45° is *n*, then the angular increment can be expressed as(1)$$\Delta =\frac{45}{n-1}\left({}^{\circ}\right)$$

The *l_i_* denotes the radial cutting length, which refers to the *i*th interpolation point. Then, the orientation and Cartesian coordinates of the *i*th interpolation point are expressed as(2)$$\begin{array}{l} \theta_{i}=\left(i-1\right)\Delta \\ x_{i}=\left(r_{0}+l_{i}\right)\cos\left(\theta_{i}\right)\\ y_{i}=\left(r_{0}+l_{i}\right)\sin\left(\theta_{i}\right) \end{array}$$

According to the principle of linear interpolation, the inner boundary after cutting can be expressed as(3)$$y=\frac{y_{i}-y_{i-1}}{x_{i}-x_{i-1}}\left(x-x_{i-1}\right)+y_{i-1}$$

In summary, the inner boundary after cutting can be controlled by the radial cutting length of interpolation points. Because the inner boundary after cutting changes the stiffness and mass of the stator, the radial cutting lengths influence the natural frequency and modal shape. In other words, the radial cutting lengths can be employed as the optimization variables.

## 3. Optimization Model

### 3.1. Optimization Model for Orthogonal Modes with Odd Nodal Diameters

The kirigami pattern guarantees intrinsic matching of orthogonal modes with odd nodal diameters, i.e., modes in which the frequencies and modal shapes of two orthogonal modes are identical, so the optimization objective only needs to minimize the distortion of the mode shape, i.e., the deviation of the modal shape from the sine-style. In order to evaluate the distortion quantitatively, the out-of-plane displacements on the circle passing through the modal shape’s peaks are extracted and then fitted to the function of “Asin(ωθ + ϕ)” using ordinary least squares. As a result, the distortion can be described by the root mean square error after the fitting. The procedure for evaluating the distortion of the modal shape is shown in Figure 3.(4)$$Distortion=\sqrt{\frac{{\sum_{i=1}^{m}\left(w_{i}-{\hat{w}}_{i}\right)}^{2}}{m}}$$
where *w_i_* is the out-of-plane displacement of the *i*th sample, *w_i_* with a hat is the one predicted by the “sine” model, and *m* is the number of samples.

The distortion is the objective function, because the more minor distortion means that the modal shape is closer to the sine-style. Meanwhile, from the results in Section 2, it is known that the radial cutting lengths influence the modal shape, so the distortion must vary with the radial cutting lengths. In summary, the optimization model can be expressed as(5)$$\begin{array}{c} Minimize\;\;Distortion\left(l_{1},\cdots ,l_{n}\right)\\ \begin{array}{cc} s.t.\;\;\; & 0\leq l_{1}\leq l_{\max}\\ & \vdots \\ & 0\leq l_{n}\leq l_{\max} \end{array} \end{array}$$
where *l_max_* is the upper boundary for the radial cutting lengths.

### 3.2. Optimization Model for Orthogonal Modes with Even Nodal Diameters

Orthogonal modes with even nodal diameters do not have intrinsic matching, i.e., the frequencies and modal shapes of the two orthogonal modes are distinct. In other words, the distortion of the modal shape and frequency difference needs to be minimized. This work defines the objective function as the weighted sum of the shape distortion and the frequency difference.(6)$$\begin{array}{c} Minimize\;\;\left(\beta_{f}\frac{f_{h}-f_{l}}{f_{h}}+\beta_{l}Distortion_{l}+\beta_{h}Distortion_{h}\right)\\ \begin{array}{cc} s.t. & 0\leq l_{1}\leq l_{\max}\\ & \vdots \\ & 0\leq l_{n}\leq l_{\max} \end{array} \end{array}$$
where *f_h_* and *f_l_* represent the lower and higher frequencies, respectively, and *Distortion_l_* and *Distortion_h_* represent the shape distortions with the lower and higher frequencies, respectively. The role of the weight coefficients *β_f_*, *β_l_*, and *β_h_* is to ensure that the three parts are in the same order.

### 3.3. Algorithm for Searching for the Solution

The computing procedure of the distortion is based on the finite element method, so the objective function does not have a closed-form derivative. Thus, an extended Nelder–Mead Simplex Algorithm with the advantages of derivative-free and bound constraints was employed. This extended Nelder–Mead Simplex Algorithm implemented lower and upper bound constraints by the careful use of transformations of the variables [27].(7)$$L_{i}=LB_{i}+\frac{\left(UB_{i}-LB_{i}\right)\left(sinz_{i}+1\right)}{2}$$
where *L_i_
*denotes the *i*th variable with a lower bound *LB_i_* and upper bound *UB_i_*. Using the transformation given in Equation (7), the new variable, *z_i_*, is no longer restricted by the bounds. As a result, the conventional Nelder–Mead Simplex Algorithm can be used to search for the solution.

However, to avoid the local optimum, the Monte Carlo Algorithm is first employed to search for a good initial value of the optimization variable for the Nelder–Mead Simplex Algorithm.

Use the Monte Carlo algorithm to compute a set of suitable *l_i_* variable and bring them into the subsequent computation. Combine Equations (2) and (7) to convert the bounded variable *l_i_* into the unbounded constraint point *z_i_*.(8)$$l_{i}=\frac{\left(l_{\max}-r_{0}\right)\left(\sin z_{i}+1\right)}{2}$$

*l_max_
*denotes the radial cutting lengths, and *r_o_
*denotes the radius of the center of the cuttings. *l_i_
*has a corresponding *z_i_*; for example, *l*_1_ corresponds to *z*_1_, *l_2_* corresponds to *z_2_*, and so on.

When establishing models, *z_i_
*needs to be converted into *l_i_*.(9)$$z_{i}=\arcsin\left(\frac{2l_{i}}{l_{\max}-r_{0}}-1\right)$$

Taking the optimization procedure of orthogonal modes with odd nodal diameters as an example, the actual steps are as follows:

Step 1: Initialize *n z_i_’*. Generate by appropriately shifting a certain step length. Then, replace *z_i_*, which is found using Equation (9), with *z_i_’*.(10)s.t.z1′=z1+b1 z2′=z2+b2 ⋮ zi′=zi+bi

*b_i_* denotes the step length, which is decided by the programming algorithm.

Step 2: Establish schemes. Replace *z_i_* with *z_i_’*, respectively, and bring them into the original scheme obtained in the Monto Carlo Algorithm. Obtain *n* schemes, and together with (*z*_1_*, z_2_, z_3_,…, z_n_*), there are a total of (*n* + 1) schemes.(11)s.t.Z1z1′,z2,z3,…,zn Z2z1,z2′,z3,…,zn … Znz1,z2,z3…,zn′i.e.Z1l1′,l2,l3,…,ln Z2l1,l2′,l3,…,ln … Znl1,l2,l3,…,ln′

*l_i_’* is the replaced length of *l_i_*.

Step 3: Convert the unbounded constraint point *z_i_* into the bounded variable *l_i_*.

Step 4: Compute the Cartesian coordinate values corresponding to *l_i_* and combine Equation (2) to obtain the corresponding coordinate values *x_i_* and *y_i_*.

Step 5: Establish finite element simulation models. Bring the coordinate values into the Cartesian rectangular coordinate system to determine the positions of each interpolation point and connect each point in sequence with a straight line; then, the internal boundary of the stator is determined, and this operation satisfies Equation (3). To control variables, the outer boundary 2*R* and the fixed constraint range are fixed values. Since the stator is a symmetrical figure, the modeling is based on 45° as the model basis, and eight identical fan-shaped parts with axial symmetries are established and combined into a complete circular ring.

Step 6: Conduct finite element simulation, compute the *Distortion* degree, and order. Obtain the out-of-plane displacement of the circular ring through finite element simulation. Obtain the modal peak values corresponding to each angle, fit “Asin(ωθ + ϕ”) by the least square method, and bring it into Equation (4) to compute the *Distortion*. Set the variance of the *Distortion* as the cut-off condition to judge whether the optimal solution is found.(12)$$s.t.\;\;Distortion\left(Z_{1}\right)\leq Distortion\left(Z_{2}\right)\leq \ldots \leq Distortion\left(Z_{n+1}\right)$$

The above assumes that 1st is the optimal solution, 2nd is the sub-optimal solution, and so on. *n* + 1th is the worst solution. The experiment is ordered according to the actual data, that is, the fitness has nothing to do with the value serial number.

Step 7: Convert the bounded variable *l_i_* into the unbounded constraint point *z_i_*, that is, the optimization of *l_i_* is converted to the optimization of *z_i_*.(13)$$Minimize\;\;Distortion\left(l_{1},\cdots ,l_{n}\right)=Minimize\;\;Distortion\left(z_{1},\cdots ,z_{n}\right)$$

Step 8: Compute the centroid scheme *Z_c_*. Discard the worst scheme, ranked *n* + 1th, and compute the average value of each parameter of the first *n* groups of schemes:(14)$$Z_{c}=\frac{1}{n}\sum_{i=1}^{n}\left(Z_{i}\right)$$

Suppose that (*z*_1_, *z*_2_, *z*_3_,…, *z_n_*) is the worst scheme; then, the centroid scheme *Z_c_* is(15)$$s.t.\;\;Z_{c}\left(\frac{{z_{1}}^{\prime }+\left(n-1\right)z_{1}}{n},\frac{{z_{2}}^{\prime }+\left(n-1\right)z_{2}}{n},\ldots ,\frac{{z_{n}}^{\prime }+\left(n-1\right)z_{n}}{n}\right)$$

Step 9: Compute the reflection scheme *Z_r_*. Choose the worst scheme, compute the reflection points of each parameter, form the reflection scheme *Z_r_*, compute the *Distortion* degree, and order.(16)$$Z_{r}=\left(1+\rho \right)Z_{c}-Z_{n+1}$$

ρ is the reflection coefficient.

Suppose that (*z*_1_*, z_2_, z_3_,…, z_n_*) is the worst scheme; then, the centroid scheme *Z_r_* is(17)$$s.t.\;\;Z_{r}\left(\left(1+\rho \right)\times \frac{{z_{1}}^{\prime }+\left(n-1\right)z_{1}}{n}-z_{1},\left(1+\rho \right)\times \frac{{z_{2}}^{\prime }+\left(n-1\right)z_{2}}{n}-z_{2},\ldots ,\left(1+\rho \right)\times \frac{{z_{n}}^{\prime }+\left(n-1\right)z_{n}}{n}-z_{n}\right)$$

Step 10: Compute the expansion scheme *Z_e_*. If *Z_r_* is better than *Z*_1_, compute the expansion points of each parameter, form the expansion scheme *Z_e_*, compute the *Distortion* degree, and order.(18)$$Z_{e}=\left(1-\beta \right)Z_{c}+\beta Z_{r}$$

β is the expansion coefficient.

Suppose that (*z*_1_*’, z_2_, z_3_,…, z_n_*) is the best scheme; then, the expansion scheme *Z_e_* is(19)$$s.t.\;\;Z_{e}\left(\left(1-\beta \right)\times \frac{{z_{1}}^{\prime }+\left(n-1\right)z_{1}}{n}+\beta \left(\left(1+\rho \right)\times \frac{{z_{1}}^{\prime }+\left(n-1\right)z_{1}}{n}-z_{1}\right),\ldots ,\left(1-\beta \right)\times \frac{{z_{n}}^{\prime }+\left(n-1\right)z_{n}}{n}+\beta \left(\left(1+\rho \right)\times \frac{{z_{n}}^{\prime }+\left(n-1\right)z_{n}}{n}-z_{n}\right)\right)$$

Step 11: Compute the contraction schemes *Z_oc_* and *Z_ic_*. If *Z_r_* is worse than *Z_n_* but better than *Z_n+_*_1_, compute the outer contraction points of each parameter, form the outer contraction scheme *Z_oc_*, compute the *Distortion* degree, and order.(20)$$Z_{oc}=\left(1-\alpha \right)Z_{n+1}+\alpha Z_{r}$$

α is the contraction coefficient.

Still suppose that (*z*_1_*, z_2_, z_3_,…, z_n_*) is the worst scheme; then, the outer contraction scheme *Z_oc_* is(21)$$s.t.\;\;Z_{oc}\left(\left(1-\alpha \right)z_{1}+\alpha \left(\left(1+\rho \right)\times \frac{{z_{1}}^{\prime }+\left(n-1\right)z_{1}}{n}-z_{1}\right),\ldots ,\left(1-\alpha \right)z_{n}+\alpha \left(\left(1+\rho \right)\times \frac{{z_{n}}^{\prime }+\left(n-1\right)z_{n}}{n}-z_{n}\right)\right)$$

If *Z_oc_* is worse than *Z_r_*, compute the inner contraction points of each parameter, form the inner contraction scheme *Z_ic_*, compute the distortion degree, and order.(22)$$Z_{ic}=\left(1-\alpha \right)Z_{c}+\alpha Z_{n+1}$$

Still suppose that (*z*_1_*, z_2_, z_3_,…, z_n_*) is the worst scheme; then, the inner contraction scheme *Z_ic_* is(23)$$s.t.\;\;Z_{ic}\left(\left(1-\alpha \right)\times \left(\frac{{z_{1}}^{\prime }+\left(n-1\right)z_{1}}{n}\right)+\alpha z_{1},\ldots ,\left(1-\alpha \right)\times \left(\frac{{z_{n}}^{\prime }+\left(n-1\right)z_{n}}{n}\right)+\alpha z_{n}\right)$$

Step 12: Compute the shrink scheme *Z_i_*. If the conditions cannot be satisfied after contraction, compute the shrink points of each parameter, form the shrink scheme *Z_i_*, compute the distortion degree, and order.(24)$$Z_{i}=\left(1-\sigma \right)Z_{1}+\sigma Z_{i}$$

σ is the shrink coefficient.

Still suppose that (*z*_1_*’, z_2_, z_3_,…, z_n_*) is the best scheme; then, the shrink scheme *Z_i_* is(25)s.t.Z21−σz1′+σz1,1−σz2+σz2′,…,1−σzn+σzn Z31−σz1′+σz1,1−σz2+σz2,…,1−σzn+σzn … Zn+11−σz1′+σz1,1−σz2+σz2,…,1−σzn+σzn

If convergence cannot be achieved after the specified number of iterations, simply select the best scheme. The schematic diagram of the optimization process is shown in Figure 4.

## 4. Optimization Result and Discussion

The material properties in the finite element model were set according to the data in Table 1, which were obtained from the literature [28,29]. The outer and initial inner radii were 2000 and 500 μm, respectively. The thicknesses of the SCS and PZT were 25 and 3 μm, respectively. The upper boundary for the radial cutting lengths *l_max_* was set to 300 μm to ensure that the kirigami pattern could effectively adjust the stator stiffness and mass. Meanwhile, the kirigami pattern will not break the peak region of the mode shape.

The simulation data were processed according to the formula derived from Section 3, and the characteristic frequency and distortion of each model were compared.

The optimized results of the orthotropic modes with odd nodal diameters are shown in Figure 5. By comparing the simulation results before and after the optimization, it can be seen that the mode shape is much closer to the sine-style following the optimization. For instance, the distortion of the B13 mode is significantly reduced from 0.055 to 0.003. Meanwhile, the B13 and B15 modes still retain the intrinsic matching after the optimization, i.e., the two orthotropic modes have the same frequency and mode shape except for a space shift. This feature accords with the deduction in Section 2, because the kirigami pattern ensures that the stiffness and mass repeat every 90 degrees.

The optimized results of the orthotropic modes with even nodal diameters are shown in Figure 6. By comparing the simulation results before and after the optimization, it can be seen that both the frequency splitting and shape distortion are suppressed. For instance, the frequency splitting relating to the B14 mode is significantly reduced from 380 Hz to 1 Hz by the optimization, and the distortions of two mode shapes are as low as 0.004. Additionally, it can be seen from Figure 6b that the mode with more nodal diameters is less improved than the one with fewer nodal diameters. For instance, the distortions of the B16 mode shape do not achieve a dramatic reduction. This phenomenon can be explained by the average elasticity in wavelength. It can be seen from Figure 7 that there is a significant difference in wavelengths between B13 and B16, and the wavelength of B13 is longer than that of B16. In the same range of 360°, there are three wavelengths for B13, while there are six wavelengths for B16. Due to the anisotropy of the material, each waveband is affected to different extents, which further influences the deformation and stress conditions of the stator. Specifically, it is manifested as the generation of different average elasticities, that is, the non-uniform deformation of the stator. Furthermore, it can be seen from Figure 7a that the elasticity at the nodal diameter position of the stator is the smallest, while it is the largest at the wave crest. Figure 7b conveys the same phenomenon. However, each waveband of B16 changes more frequently, which makes the wave crest and wave trough more obviously affected by the anisotropy of the material; that is, different vibration modes of the stator are exhibited. As a result, in comparison to the mode shape with fewer wavelengths, the average elasticity at any two wavelengths must be more different from each other than the mode shape is with other wavelengths. As such, the mode shape with more wavelengths must be more difficult to modify.

In engineering, fewer wavelengths can result in fewer electrodes and electrical contacts. The stator with fewer wavelengths is more flexible, so the driving voltage is also lower.

Thus, modes with fewer wavelengths, such as B13 and B14, are usually the first selection. Mode optimization based on kirigami can realize the objective of suppressing the mode imperfections that are induced by elasticity.

## 5. Conclusions

In this study, mode optimization based on kirigami is studied to suppress modal splitting and shape distortion and realize high-quality traveling waves. The proposed kirigami pattern, built using linear interpolation, retained the intrinsic matching of orthogonal modes with odd nodal diameters. Thus, the objective function for orthogonal modes with odd nodal diameters only contained the shape distortion. However, the objective function for orthogonal modes with even nodal diameters contained both shape distortion and modal splitting. After optimization, the mode shape of the orthogonal modes with odd nodal diameters is much closer to the sine-style. For instance, the distortion of the B13 mode is significantly reduced to 0.003. On the other hand, the optimization suppresses both frequency splitting and shape distortion for orthogonal modes with even nodal diameters. For instance, the frequency splitting referring to the B14 mode is significantly reduced from 380 Hz to 1 Hz, and the shape distortion is as low as 0.004.

In the future, a more complex kirigami pattern based on topological optimization can be studied to acquire a more perfect mode for MEMS TWUMs.

## Figures and Tables

**Figure 1 micromachines-16-00239-f001:**
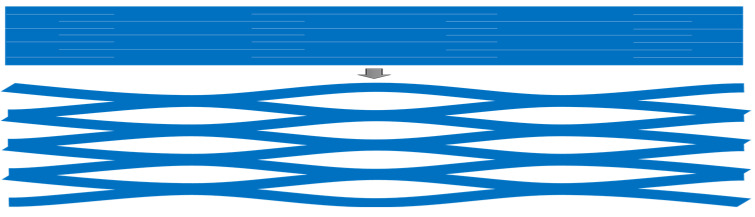
Stretched patterns processed by kirigami.

**Figure 2 micromachines-16-00239-f002:**
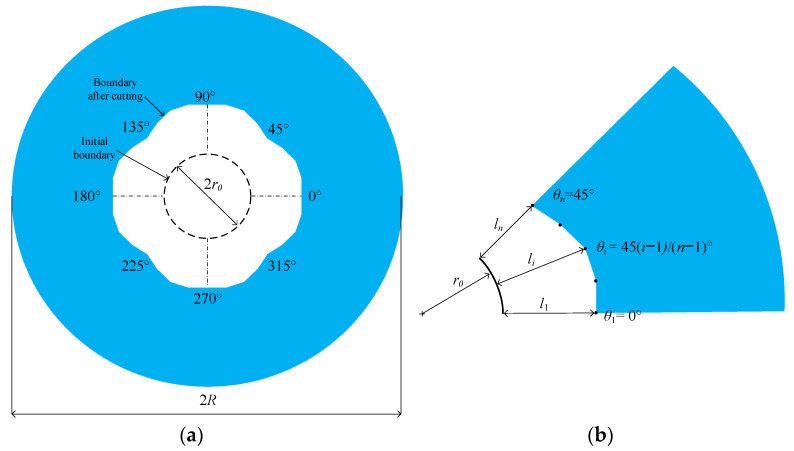
A schematic diagram of the kirigami pattern. *l_i_* denotes the length of the *i*th cutting, *n* denotes the number of the cuttings, *r_o_* denotes the radius of the center of the cuttings, *l_n_* and *l*_1_ denote the left and right ends of the first cutting, respectively, and *R* represents the radius of the anchor. (**a**) Internal boundary after processing by kirigami (**b**) Internal boundary formed by connecting interpolation points in the range of 0° to 45°.

**Figure 3 micromachines-16-00239-f003:**
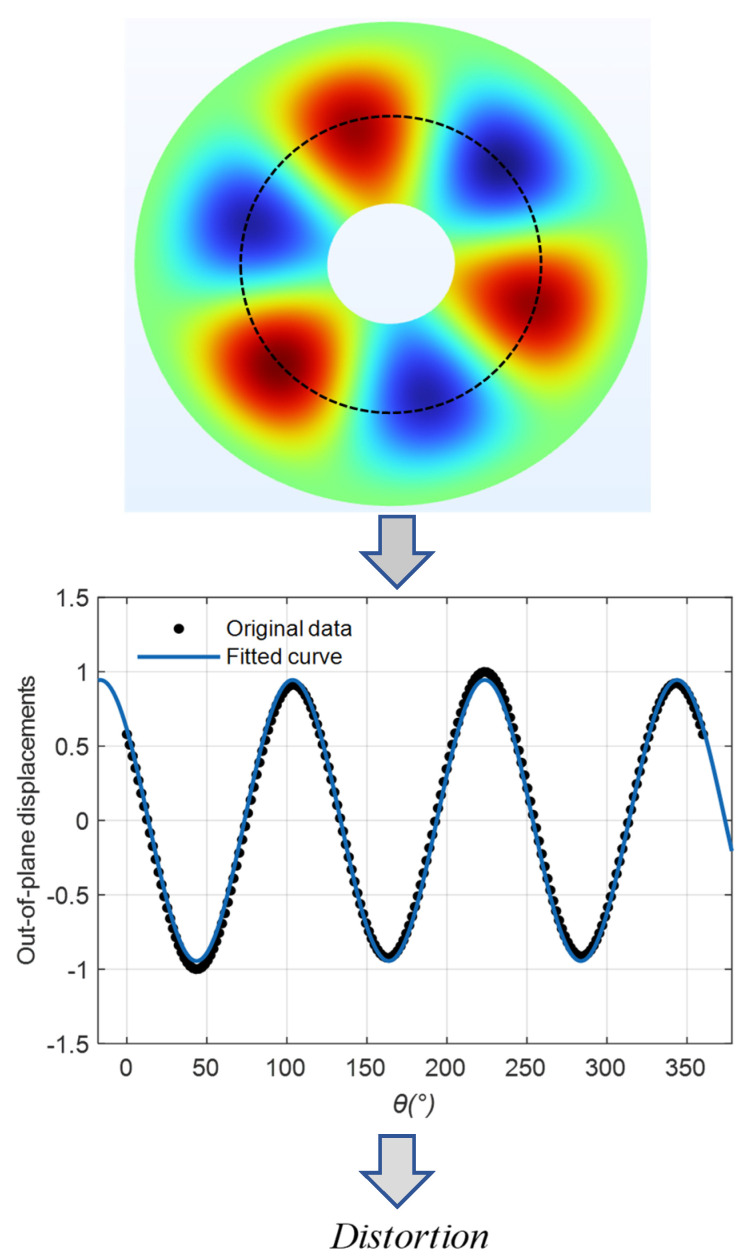
The procedure for evaluating the distortion of the modal shape.

**Figure 4 micromachines-16-00239-f004:**
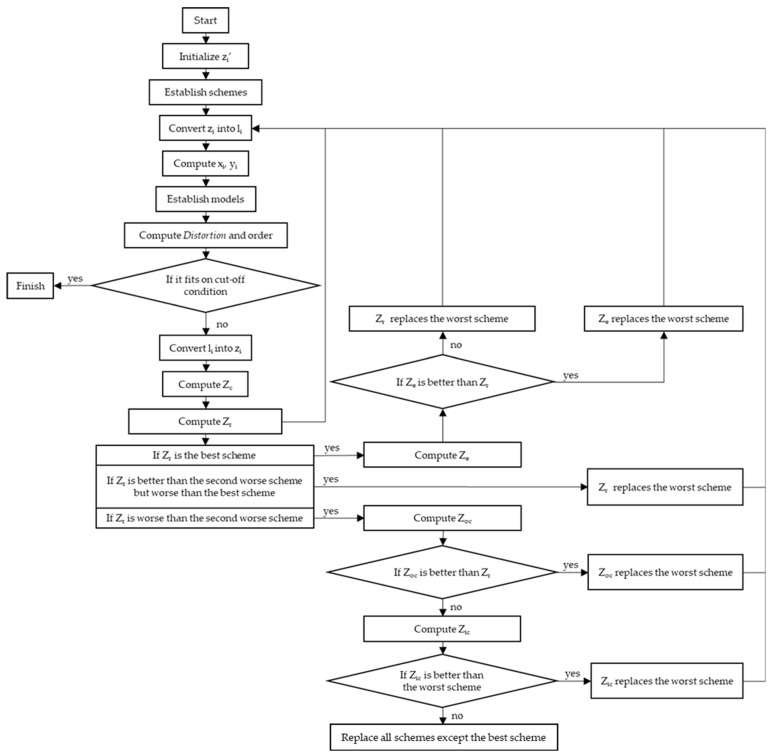
The process of the Nelder–Mead Simplex Algorithm for finding the optimal solution.

**Figure 5 micromachines-16-00239-f005:**
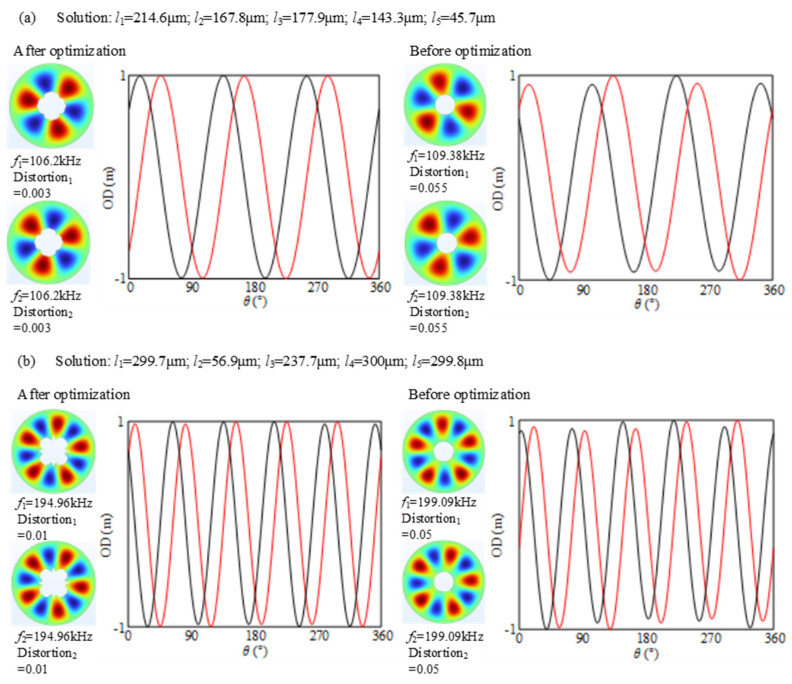
Optimized results of the orthotropic modes with odd nodal diameters: (**a**) optimized results in B13 mode; (**b**) optimized results in B15 mode. The out-of-plane displacements on the peaking circle are extracted to describe the shape distortion. The red and dark lines represent 1st and 2nd mode shapes, respectively.

**Figure 6 micromachines-16-00239-f006:**
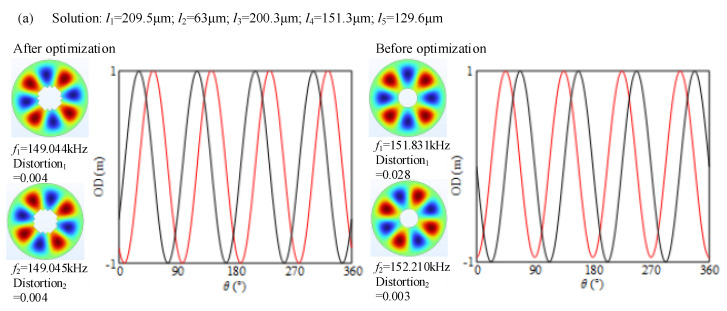
Optimized results of the orthotropic modes with even nodal diameters: (**a**) optimized results in B14 mode; (**b**) optimized results in B16 mode. The out-of-plane displacements on the peaking circle are extracted to describe the shape distortion. The red and dark lines represent 1st and 2nd mode shapes, respectively.

**Figure 7 micromachines-16-00239-f007:**
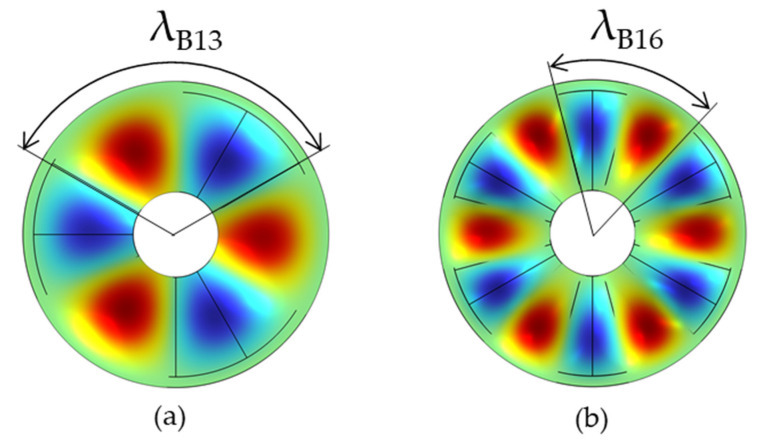
Wavelengths vary with different nodal diameters: (**a**) optimized results in B13 mode; (**b**) optimized results in B16 mode.

**Table 1 micromachines-16-00239-t001:** Material properties and dimensions.

	PZT	SCS
Density *ρ* (kg/m^3^)	7800	2330
Elastic constants *C*_11_, *C*_22_ (GPa)	132	194.5
Elastic constants *C*_12_, *C*_21_ (GPa)	71	35.7
Elastic constants *C*_13_, *C*_31_, *C*_23_, *C*_32_ (GPa)	73	64.1
Elastic constants *C*_33_ (GPa)	115	165.7
Elastic constants *C*_44_, *C*_55_ (GPa)	30	79.6
Elastic constants *C*_66_ (GPa)	26	50.9
Other elastic constants (GPa)	0	0
Thickness(μm)	3	25
Outer radius *R* (μm)	2000	2000
Initial inner radius *r_0_* (μm)	500	500
Upper boundary for the radial cutting lengths *l_max_* (μm)	300	300

## Data Availability

Data are contained within the article.

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
