# Peer review of "Mode Optimization of Microelectromechanical-System Traveling-Wave Ultrasonic Motor Based on Kirigami"

_micromachines, 2025, doi:10.3390/mi16020239_

Round 1

Reviewer 1 Report

Comments and Suggestions for Authors

This work introduced an efficient method to realize mode optimization. The idea is novel and is supported by the simulation results. The following issues should be detailed.

1. How the upper boundary for the radial cutting length was set?

2. Is the symmetry of the finite element model a factor of mode distortion? Will the size of meshing grid affects the distortion defined?

Author Response

Comments 1: How the upper boundary for the radial cutting length was set?

Response 1: Acquired by experience. Usually, the size is not fixed, so we use the size of our lab's previous designs to carry out the experience value.

Comments 2: Is the symmetry of the finite element model a factor of mode distortion? Will the size of meshing grid affects the distortion defined?

Response 2: Thank you for pointing out those comments. We agree that the finite element software does have computational errors. However, the calculation error in this article is small, so it can be ignored. For the first question, symmetry for the smaller models in the calculation will indeed reduce the accumulation of errors and, to a certain extent, reduce the possibility of modal distortion. However, there may be asymmetrical errors in actual manufacturing, which will affect the modal results. Therefore, the experimental models in this article all use the ideal form of symmetrical structure to ensure the consistency of comparison. The effect of symmetry on the mode can be further discussed in future research. For the second question, a smaller mesh size is used to simulate the model, which improves the accuracy of the calculation results.

Reviewer 2 Report

Comments and Suggestions for Authors

- The approach of restoring symmetry through Kirigami patterns to address mode distortion caused by the anisotropic elasticity of single-crystal silicon used in MEMS TWUM is effective.

- It is necessary to provide sufficient background knowledge in the introduction.

- In '2. Kirigami Pattern on Inner Boundary', Equations (1) to (4) involve straightforward mathematical derivations and can be condensed for brevity.

- Even with the optimization of the Kirigami pattern, mode distortion caused by the difficulty in achieving machining precision for the Kirigami compared to the original cylindrical section needs to be validated through experimental results.

- Even if the Kirigami pattern minimizes mode distortion, it is necessary to consider whether any performance degradation or other issues arise in the TWUM.

Comments on the Quality of English Language

- To be improved.

Author Response

Comments 1: The approach of restoring symmetry through Kirigami patterns to address mode distortion caused by the anisotropic elasticity of single-crystal silicon used in MEMS TWUM is effective.

Response 1: Thanks for your affirmation.

Comments 2: It is necessary to provide sufficient background knowledge in the introduction.

Response 2: Agree. I have reorganized the introduction content and detailed the recent scholars' research on traveling wave distortion. Please see line numbers 45-66 for details.

Comments 3: In '2. Kirigami Pattern on Inner Boundary', Equations (1) to (4) involve straightforward mathematical derivations and can be condensed for brevity.

Response 3: Thank you for pointing this out, but I thought these steps would explain the calculation method in detail to readers and illustrate the meaning of the interpolation points. Please pardon me for that I didn’t change those steps.

Comments 4: Even with the optimization of the Kirigami pattern, mode distortion caused by the difficulty in achieving machining precision for the Kirigami compared to the original cylindrical section needs to be validated through experimental results.

Response 4: Thank you very much for your valuable suggestions. As you pointed out, we fully recognize the importance of verifying the theory through experiments to improve the quality and persuasiveness of the article. At present, however, we do face practical difficulties that prevent this experiment from being carried out immediately. On the one hand, the schedule is very tight, and it is really difficult to spare enough time to conduct new experiments shortly. On the other hand, some of the equipment in our laboratory cannot meet the requirements of this experiment for the time being, and applying for new equipment or using external equipment requires a certain process and time to coordinate. Nevertheless, our existing research results and relevant data have been able to support the theory proposed in the paper to a certain extent. At the same time, we are also aware of the indispensability of experimental verification and will make detailed plans to carry out this experiment in the future.

Comments 5: Even if the Kirigami pattern minimizes mode distortion, it is necessary to consider whether any performance degradation or other issues arise in the TWUM.

Response 5: I agree with your point of view, and we will continue to explore the performance impact of TWUM under the technology of kirigami in the following research.

Reviewer 3 Report

Comments and Suggestions for Authors

The article is devoted to optimization of traveling wave ultrasonic mems, where stator performance characteristics are improved through the use of the kirigami technique. The article is generally good. The idea of ​​the Kirigami technique for the task at hand is described in sufficient detail. The introduction contains enough information to understand the basics of the problem under consideration and the current state of the issue. The bibliography includes mainly recent works. The originality of the work is 74%.  Much of the introduction is taken from the previous work of the authors https://doi.org/10.1016/j.ymssp.2022.110083.

However, some not entirely clear points remain.

What FEA software was used to calculate the displacements?

In what language was the described optimization algorithm implemented?

How many iterations and time did it take to solve one problem?

Did the calculation take place within the framework of linear electroelasticity?

Were any scale effects taken into account, such as those associated with the gradient theory of electroelasticity? Still, the dimensions of the device are quite small, especially the thickness.

Please explain the term on line 269 “average elasticity in wavelength”.

Author Response

Comments 1: What FEA software was used to calculate the displacements?

Response 1: COMSOL.

Comments 2: In what language was the described optimization algorithm implemented?

Response 2: The MATLAB algorithm is used to calculate the result.

Comments 3: How many iterations and time did it take to solve one problem?

Response 3: The Monte Carlo Algorithm is iterated 500 times, and the Nelder-Mead Simplex Algorithm is iterated 100 times, which takes a total of 3 hours.

Comments 4: Did the calculation take place within the framework of linear electroelasticity?

Response 4: Electroelasticity refers to the piezoelectric characteristics, the simulation optimization of this experiment is the mode, only the use of solid mechanics for modal analysis, did not involve the transient analysis after adding voltage. So this paper does not involve the problem of electroelasticity.

Comments 5: Were any scale effects taken into account, such as those associated with the gradient theory of electroelasticity? Still, the dimensions of the device are quite small, especially the thickness.

Response 5: As in problem 2, there is no electroelasticity involved.

Comments 6: Please explain the term on line 269 “average elasticity in wavelength”.

Response 6: The average elasticity of materials refers to the average performance of elastic deformation behavior of materials under certain conditions when subjected to external forces. It comprehensively considers the elastic characteristics of materials under different force directions, different positions and different load conditions, and is a generalization of the overall elastic properties of materials. As can be seen from Figure 7, different color distributions represent different elasticity, and a complete wavelength is a sector area. For example, the wavelength region of B13 is compared with that of B16, and the elasticity of B16 changes greatly, which is reflected in the obvious color change.

Round 2

Reviewer 2 Report

Comments and Suggestions for Authors

While I understand the difficulties related to scheduling and equipment issues preventing the experiment, it is important to find ways to address these challenges in order to enhance the credibility and academic value of the theory. Currently, it is difficult to fully establish the reliability of the results presented without experimental verification. Moreover, considering that previous studies have advanced their theories based on experimental results, it is essential to prioritize finding a way to conduct the experiment. Striking a balance between theoretical contributions and experimental verification is crucial, and it is important to recognize that theories presented without experimental validation may be limited in their applicability.

Author Response

Comments 1: While I understand the difficulties related to scheduling and equipment issues preventing the experiment, it is important to find ways to address these challenges in order to enhance the credibility and academic value of the theory. Currently, it is difficult to fully establish the reliability of the results presented without experimental verification. Moreover, considering that previous studies have advanced their theories based on experimental results, it is essential to prioritize finding a way to conduct the experiment. Striking a balance between theoretical contributions and experimental verification is crucial, and it is important to recognize that theories presented without experimental validation may be limited in their applicability.

Response 1: I still agree with you. When we have the funding, the experiment will be started, and I hope you understand. I have added more details of the steps in “3.3. Algorithm for Search for the Solution”, which are enriched to clarify the optimization idea. Formulas 8-25 specify the logic of the algorithm. In the discussion, the effect of wavelength on the average elasticity is explained in more detail, and Figure 7 is added for its analysis. This recomposition can be seen from the 269th to 283th lines.
